# A Novel Contrastive Learning Method for Clickbait Detection on RoCliCo: A Romanian Clickbait Corpus of News Articles

**Daria-Mihaela Broscoțeanu**
Department of Computer Science
University of Bucharest
Romania
dariabroscoteanu@gmail.com

**Radu Tudor Ionescu**
Department of Computer Science
University of Bucharest
Romania
raducu.ionescu@gmail.com

## Abstract

To increase revenue, news websites often resort to using deceptive news titles, luring users into clicking on the title and reading the full news. Clickbait detection is the task that aims to automatically detect this form of false advertisement and avoid wasting the precious time of online users. Despite the importance of the task, to the best of our knowledge, there is no publicly available clickbait corpus for the Romanian language. To this end, we introduce a novel Romanian Clickbait Corpus (RoCliCo) comprising 8,313 news samples which are manually annotated with clickbait and non-clickbait labels. Furthermore, we conduct experiments with four machine learning methods, ranging from handcrafted models to recurrent and transformer-based neural networks, to establish a line-up of competitive baselines. We also carry out experiments with a weighted voting ensemble. Among the considered baselines, we propose a novel BERT-based contrastive learning model that learns to encode news titles and contents into a deep metric space such that titles and contents of non-clickbait news have high cosine similarity, while titles and contents of clickbait news have low cosine similarity. Our data set and code to reproduce the baselines are publicly available for download at https://github.com/dariabroscoteanu/RoCliCo.

## 1 Introduction

Clickbait detection (Biyani et al., 2016; Potthast et al., 2016), the task of automatically identifying misleading news titles, has received a raising interest in recent years (Dong et al., 2019; Fakhruzzaman et al., 2021; Indurthi et al., 2020; Kumar et al., 2018; Liu et al., 2022; Pujahari and Sisodia, 2021; Rony et al., 2018; Vorakitphan et al., 2019; Wu et al., 2020; Yi et al., 2022) due to its importance in addressing concerns regarding the deceptive behavior of online media platforms aiming to increase their readership and revenue in an unethical manner. The task has been widely explored in multiple languages, such as English (Agrawal, 2016;

Potthast et al., 2016, 2018; Pujahari and Sisodia, 2021; Wu et al., 2020; Yi et al., 2022), Chinese (Liu et al., 2022), Hindi (Kaushal and Vemuri, 2020), and Turkish (Genç and Surer, 2023), among others. However, for low-resource languages, such as Romanian, there is a comparatively lower amount of research studying clickbait detection (Fakhruzzaman et al., 2021; Vincze and Szabó, 2020), primarily due to the lack of annotated resources. In this context, we introduce a novel **Ro**manian **Cli**ckbait **Co**rpus (RoCliCo) comprising 8,313 news samples which are manually annotated with clickbait and non-clickbait labels. To the best of our knowledge, RoCliCo is the first publicly available corpus for Romanian clickbait detection. The news articles were collected from six publicly available news websites from Romania, while avoiding overlapping publication sources between our training and test splits. This ensures that the reported performance levels are not artificially increased because of models relying on factors such as authorship or publication source identification.

It is important to clearly separate between satirical, fake and clickbait news. Satirical news represents a type of fake news, where the intention of the author is to make the readers laugh. In satirical news, it is clear to the readers that the reported events are fictitious. Fake news reports fake events or facts, where the intention of the author is to deceive the readers into believing that the fake events or facts are true. Clickbait news represents a distinct category where the news title is deceptive with respect to the content. In some clickbait news, the content can present real events and true facts. In other clickbait news, the reported events or facts can be fake. Hence, these three categories (satirical, fake, clickbait) are distinct, although their intersection is not necessarily empty.

We perform experiments with several machine learning models. Because some recent challenge reports (Chakravarthi et al., 2021) showed that shal-

low and handcrafted models can sometimes surpass deep models, we include both handcrafted and deep models as baselines for RoCliCo. Hence, two baselines are based on handcrafted features that are commonly used for clickbait detection, while the other three are based on deep learning architectures, such as a Long Short-Term Memory (LSTM) network (Hochreiter and Schmidhuber, 1997), and two versions of Bidirectional Encoder Representations from Transformers (BERT) (Devlin et al., 2019) for Romanian (Dumitrescu et al., 2020). Among the considered baselines, we propose a novel BERT-based contrastive learning model that learns to jointly encode news titles and contents into a deep metric space. The embedding space is constructed such that titles and contents of non-clickbait news have high cosine similarity, while titles and contents of clickbait news have low cosine similarity. Thus, the similarity between the title and the content of an article is leveraged as a clickbait indicator. Finally, the list of models evaluated on RoCliCo is completed by an ensemble of the five individual baselines, which is based on weighted voting.

In summary, our contribution is twofold:

- We introduce the first publicly available corpus of Romanian news articles comprising manually annotated clickbait and non-clickbait samples.

- We propose a novel clickbait detection model based on learning a deep metric space where both news titles and contents are jointly represented. In the shared embedding space, titles and contents of non-clickbait news are supposed to be neighbors, while titles and contents of clickbait news should be far apart.

## 2 Related Work

**Related corpora.** Clickbait detection was first studied for the English language, several English resources being available to date (Potthast et al., 2016, 2018). With the rising interest towards solving the task, researchers started to compile and publish clickbait detection corpora for various languages, including Chinese (Liu et al., 2022), Hindi (Kaushal and Vemuri, 2020), Hungarian (Vincze and Szabó, 2020), Indonesian (Fakhruzzaman et al., 2021), Thai (Klairith and Tanachutiwat, 2018) and Turkish (Genç and Surer, 2023). These resources vary in size, ranging from 180 data samples for Hungarian (Vincze and Szabó, 2020) to 52,000

data samples for Hindi (Kaushal and Vemuri, 2020). There are three corpora containing more than 30K samples, but these comprise only tweets. Corpora formed of full news articles usually have less than 10K samples. With over 8K samples, RoCliCo is similar in size with the resources available for other languages. However, most of the other corpora are based on distant supervision, labeling the news as clickbait based on the source platforms they belong to. In contrast, our corpus is based on manual annotations. We note that news articles from a certain source platform cannot be placed in the same bucket, although it is true that different platforms use clickbait in different percentages. For RoCliCo, these percentages vary between 17% and 71%, supporting our previous statement.

To the best of our knowledge, there are no publicly available data sets for Romanian clickbait detection. We thus consider RoCliCo as a useful resource for the low-resource Romanian language.

**Related methods.** To date, researchers proposed various clickbait detection methods, ranging from models based on handcrafted features (Potthast et al., 2016) to deep neural networks (Fakhruzzaman et al., 2021; Liu et al., 2022; Wu et al., 2020). However, most of these approaches are based on standard learning objectives, little attention being dedicated to alternative optimization criteria, such as those based on contrastive learning (Yi et al., 2022; López-Sánchez et al., 2018). Yi et al. (2022) trained a Variational Auto-Encoder to jointly predict labels from text and generate text from labels using a mixed contrastive loss, while López-Sánchez et al. (2018) employed contrastive learning to minimize the distances between headlines from the same category and maximize the distances between headlines from different categories. Our model learns a metric space between titles and contents, while existing contrastive learning methods learn a metric space between titles, which is fundamentally different. To the best of our knowledge, we are the first to propose a contrastive learning objective based on title and content pairs. The novelty of our approach consists of directly transforming each news article into a pair, essentially eliminating the need of performing hard example mining (Suh et al., 2019).

**Related tasks.** Although fake news and clickbait detection are sometimes studied jointly (Bourgonje et al., 2017), we would like to underline once again that fake news detection and clickbait detection are

distinct tasks, i.e. a clickbait title does not necessarily imply that the respective news article is fake. For example, an incomplete title might be sufficient to attract readers, only to reveal the full (and less attractive) story in the content, while presenting only true facts. Still, we would like to acknowledge that related tasks, namely fake (Busioc et al., 2020) and satirical (Rogoz et al., 2021) news detection, have been studied for Romanian.

## 3 Corpus

RoCliCo gathers news from six of the most prominent Romanian news websites (Cancan, Digi24, Libertatea, ProTV, WowBiz, Viva), while only containing articles from the public web domain, being free of subscription. The corpus consists of 8,313 samples distributed over two classes: 4,593 non-clickbait and 3,720 clickbait. As the corpus is manually labeled, each sample contains a title, a body, and a label that deems it clickbait or not by taking into consideration the characteristics of both text and title. There are 3,942,790 tokens in our data set, the average number of tokens per article being roughly 474. The average title length is 21 tokens, while the average content length is 454 tokens. As expected, the titles are much shorter than the contents. In terms of the number of sentences, the news articles vary between 3 and 448. The average number of sentences per news article is 28.

All data samples were labeled by annotators who received detailed instructions on the factors that should be taken into account when labeling news as clickbait or non-clickbait. The annotators are native Romanian speakers, who graduated from their bachelor studies at Romanian universities. We checked their reading comprehension skills prior to enrollment. Each annotator was asked to read an instructive document written in Romanian, where we provided definitions, guidelines and examples of clickbait and non-clickbait news articles. The labeling process, completed by three annotators, consists of filtering each sample through a set of criteria by identifying the properties of the title and the relationship between the body and the headline (title). As clickbait titles tend to misguide or attract readers by overemphasizing certain aspects, we recommended annotators to take into considerations the following characteristics: containing questions for the reader, teasers, emotionally evocative structures, suggesting the existence of a photo or video, being controversial, or being racist. We note that the title's inclusion in one of the following genres may also represent an indicator for its membership to the clickbait category: political, finance, celebrity. Consequently, we instructed the annotators to be extra careful when they have to annotate a news article from the politics, finance, or celebrity categories. These genres are picked by studying Romanian news websites and finding that the most common themes likely to contain clickbait are the ones above. To assign the non-clickbait category, even if one of the above criteria fits for the title, the body and its similarity with the headline determine the final separation line between the two classes. The final label for a sample is decided based on the majority vote of three annotators, which reduces labeling errors by a large extent. The Cohen's kappa coefficient among our annotators is 0.73, indicating a substantial agreement between annotators. We confirm the consistency of the labels by checking the assigned labels for a subset of 200 samples. After completing the labeling process, we found that the clickbait ratio per source ranges between 17% and 71%, indicating that all the considered news sources seem to use clickbait titles to some extent. This is likely because the chosen sources do not use a subscription-based publication format, and the news publishers have to rely on advertisement to obtain earnings.

We separate the news platforms between the training and test sets, keeping the news from Cancan, ProTV and WowBiz for training and those from Libertatea, Viva and Digi24 for testing. As such, the training set consists of 6,806 articles, while the test set comprises a total of 1,507 articles. The distribution of classes per subset is presented in Table 1. We underline that the training set incorporates a more even data distribution across the two classes, providing sufficient examples to represent both classes and to ensure a smooth learning process. In contrast, the test data represents a more natural distribution, with the non-clickbait class being dominant at a ratio of 2.4:1.

Without source separation between training and test, a model that overfits to certain source-specific aspects that are not related to clickbait (e.g. author style, preference towards certain subjects, and so on) will reach high scores on the test. However, these scores are unlikely to represent the actual performance of the model in a real-world scenario, where news can come from different sources without annotated samples to be used for training. Thus,

| Subset | Clickbait | Non-clickbait | Total |
|--------|-----------|---------------|-------|
| Train  | 3,279     | 3,527         | 6,806 |
| Test   | 441       | 1,066         | 1,507 |
| **Total** | **3,720** | 4,593      | 8,313 |

Table 1: Data distribution of RoCliCo.

we consider that a more realistic evaluation is to separate the platforms. In the proposed setting, models that learn patterns related to author style, preferred themes and such will not be able to capitalize on features unrelated to clickbait detection.

## 4 Baselines

**RF and SVM based on handcrafted features.** Our first baseline embodies a morphological, syntactical and pattern analysis model, similar to the recent approach of Coste et al. (2020), which uses a set of engineered features based on part-of-speech tagging, scores (CLScore, LIX, and RIX), and punctuation patterns. First, we lowercase the titles and strip down special characters. Next, we extract features such as the number of question words, punctuation patterns in the headline, the LIX and RIX indexes (Anderson, 1983), the CLScore (Coleman and Liau, 1975), part-of-speech tags based on Stanza (Qi et al., 2020), the number of common nouns, and the number of proper nouns. These features are then passed to two separate models, a Random Forest (RF) model and a Support Vector Machines (SVM) classifier.

**BiLSTM network.** Our second baseline approaches the problem via a recurrent architecture based on bidirectional LSTM (BiLSTM) layers (Graves et al., 2005). There are two BiLSTM branches, one for titles and one for contents. Each input is first preprocessed by the Keras tokenizer. Next, word embeddings are extracted using Fast-Text (Bojanowski et al., 2017). The resulting embeddings are passed as input to the BiLSTM neural network. The outputs of the separate BiLSTM branches are pooled, concatenated, and further passed through two fully-connected layers and a softmax classification layer.

**Fine-tuned Ro-BERT.** Our third baseline involves the fine-tuning of the Romanian BERT (Dumitrescu et al., 2020), a transformer-based model trained exclusively on a comprehensive collection of Romanian corpora. To fine-tune the model, the title and body of each news sample are concatenated using the separation token *[SEP]* as a delimiter, and the result is encoded using the Ro-BERT encoder.

The *[CLS]* tokens returned by the last transformer block are used as sample embeddings in our classification task. The Ro-BERT embeddings are passed through a dropout layer, a fully-connected layer, and a classification layer. Finally, the labels are predicted by applying argmax to the probabilities produced by the classification layer.

**Contrastive Ro-BERT.** As a novel technique to solve the clickbait detection problem, our contrastive learning approach employs Siamese networks (Chicco, 2021) to learn the similarity between news headlines and contents. This is achieved by harnessing the distance between headlines and contents in a loss function that penalizes the clickbait pairs that are too close or the non-clickbait pairs that are too far apart. Let $X = \{x_1, x_2, ..., x_n\}$ and $Y = \{y_1, y_2, ..., y_n\}$ represent our training set of news samples and the set of corresponding labels, where $x_i = (t_i, c_i)$ is a pair given by title $t_i$ and content $c_i$, and $y_i \in \{0, 1\}$ is the class label (0 for clickbait and 1 for non-clickbait), for all $i = \{1, 2, ..., n\}$. Our approach applies a Siamese Ro-BERT model to generate the normalized embeddings $\mathbf{v}_{t_i}$ and $\mathbf{v}_{c_i}$ for each title $t_i$ and body $c_i$, respectively. We further compute the proposed contrastive loss, as follows:

$$
\mathcal{L}_{CL} = \frac{1}{n} \sum_{i=1}^{n} (y_i \cdot \delta\left(\mathbf{v}_{t_i}, \mathbf{v}_{c_i}\right) + \\
+ (1 - y_i) \cdot \max\left(0, m - \delta\left(\mathbf{v}_{t_i}, \mathbf{v}_{c_i}\right)\right),
\tag{1}
$$

where $m$ is a margin that limits the optimization for the clickbait pairs, and $\delta$ is the cosine dissimilarity between $\mathbf{v}_{t_i}$ and $\mathbf{v}_{c_i}$, i.e.:

$$
\delta\left(\mathbf{v}_{t_i}, \mathbf{v}_{c_i}\right) = 1 - \frac{\mathbf{v}_{t_i} \cdot \mathbf{v}_{c_i}}{\|\mathbf{v}_{t_i}\| \cdot \|\mathbf{v}_{c_i}\|}.
\tag{2}
$$

We set $m = 1$ in our experiments. Unlike our approach, related contrastive methods use $(t_i, t_j)$ pairs, i.e. a pair is composed of two titles of different news articles. Such methods require hard example mining, resulting in an expensive training process. In contrast, our approach has the same training complexity as the standard BERT.

**Ensemble.** As our baselines consist of relatively distinct approaches to the problem, we consider the idea of combining the results into an ensemble of classifiers worthy of investigation. We gather the soft labels predicted by the baselines presented so far and use a weighted voting mechanism to predict the label of each sample. The weights are determined by computing the number of correct predictions generated by each model on a separate

| Model | Clickbait | | | Non-Clickbait | | | Macro |
|---|---|---|---|---|---|---|---|
| | Precision | Recall | F$_1$ Score | Precision | Recall | F$_1$ Score | F$_1$ Score |
| Random Forest | 0.8666 | 0.6190 | 0.7222 | 0.8590 | 0.9606 | 0.9070 | 0.8146 |
| SVM | 0.8607 | 0.7006 | 0.7725 | 0.8850 | 0.9531 | 0.9178 | 0.8451 |
| BiLSTM | 0.7485 | 0.8367 | 0.7901 | 0.9290 | 0.8837 | 0.9058 | 0.8495 |
| Fine-tuned Ro-BERT | 0.9436 | 0.7210 | 0.8174 | 0.8948 | 0.9821 | 0.9364 | 0.8769 |
| Contrastive Ro-BERT | 0.9153 | 0.8571 | 0.8852 | 0.9424 | 0.9672 | 0.9546 | 0.9199 |
| Ensemble | 0.8981 | 0.8390 | 0.8675 | 0.9352 | 0.9606 | 0.9477 | 0.9076 |

Table 2: Precision, recall and F$_1$ scores of the proposed baselines on the official RoCliCo test set. The scores are independently reported for each class to enable a detailed class-based assessment of detection performance.

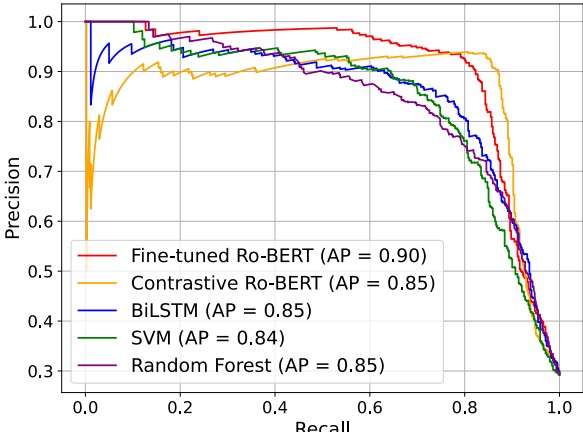

Figure 1: Precision-recall (PR) curves of the individual models evaluated on RoCliCo. The Average Precision (AP) of each model is reported in the legend, next to each model's name. Best viewed in color.

validation set, which is distinct from the test data. The weight of a model is equal to the ratio between the number of correct labels and the number of validation samples.

**Reproducibility note.** Appendix A provides details about optimal hyperparameter values.

## 5 Results

In our empirical study, we perform binary classification experiments aimed at predicting whether a news is clickbait or non-clickbait. As evaluation metrics, we report the precision, recall and F$_1$ score for each of the two classes, on the official test set.

The results obtained by the individual baselines, as well as their weighted ensemble, are shown in Table 2. Among the individual baselines, the LSTM network and our contrastive learning approach achieve the highest recall rates on clickbait news. The highest precision is obtained by the fine-tuned Ro-BERT, but our contrastive version reaches the highest F$_1$ score (0.8852) on clickbait articles. On non-clickbait news, the best models are the fine-tuned Ro-BERT and the contrastive

Ro-BERT, the latter model reaching the highest F$_1$ score of 0.9546.

We evaluate the statistical significance of the proposed contrastive Ro-BERT with respect to the fine-tuned Ro-BERT using a paired McNemar's test. The test indicates that the performance improvement brought by our contrastive learning method is statistically significant, with a p-value of 0.001.

In Figure 1, we illustrate the precision-recall (PR) curves of the individual models on the test set. Each curve is summarized via the Average Precision (AP) measure, which represents the area under the PR curve. In terms of AP, it looks like the best model is the fine-tuned Ro-BERT. Although the AP score of our contrastive Ro-BERT model is lower, our approach generates the sweetest spot on the graph, i.e. the closest point to the top-right corner. This point corresponds to the best trade-off between precision and recall.

The overall results of the models evaluated on the clickbait detection task suggest that predicting non-clickbait articles is fairly easy. However, clickbait articles are comparatively harder to detect, regardless of the employed model. This indicates that Romanian clickbait detection is a challenging task, which remains to be solved in the future with the help of our new corpus.

## 6 Conclusion

In this work, we introduced RoCliCo, the first public corpus for Romanian clickbait detection. We trained and evaluated four models and an ensemble of these models to establish competitive baselines for future research on our corpus. Among these models, we proposed a novel approach, namely a BERT model based on contrastive learning that uses the cosine similarity between title and content to detect clickbait articles. The empirical results indicate that there is sufficient room for future research on the Romanian clickbait detection task.

## 7 Limitations

This work is fully focused on our Romanian clickbait detection data set, but the performance levels of the considered approaches might be different on other languages. Due to our specific focus on the Romanian language, we did not test the generalization capacity of the considered models across multiple languages. This could be addressed in future work, in an article with a broader focus.

Another limitation of our work is the rather limited number of samples included in our corpus. Our examples are collected from six of the most common Romanian news websites, but there are several other websites of Romanian news that have not been considered. Extending the list of news sources should definitely result in collecting more news articles. However, the limitation is not caused by the low number of chosen websites, but by the laborious annotation process involving human effort. Our annotation process is based on native Romanian volunteers who understood the importance of studying and solving the clickbait detection task, and agreed to label the collected articles for free. Labeling more samples would require the involvement of more annotators, but we were unable to find them due to the lack of funding for this project.

## 8 Ethics Statement

The data was collected from six Romanian news websites. The news articles are freely accessible to the public without any type of subscription. As the data was collected from public websites, we adhere to the European regulations[1] that allow researchers to use data in the public web domain for non-commercial research purposes. We thus release our corpus as open-source under a non-commercial share-alike license agreement.

The manual labeling was carried out by volunteers who agreed to annotate the news articles for free. Prior to the annotation, they also agreed to let us publish the labels along with the data set.

Our corpus is collected from five different websites, so it may include linguistic variations, such as different styles, terminology, and so on. Information about the authors, such as gender, age or cultural background, is not included in our corpus to preserve the anonymity of the authors.

We acknowledge that some news samples could refer to certain people, e.g. public figures in Roma-

nia. Following GDPR regulations, we will remove all references to a person, upon receiving removal requests via an email to any of the authors.

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

## A  Appendix: Models and Parameters

This appendix serves as a comprehensive overview of the architectural design and the specific hyperparameters used for each model, to improve reproducibility and transparency. This appendix also contains supplementary information about the choices made during the experimental setup. Aside from the details given below, we release the code to reproduce the baselines along with our corpus.

**RF and SVM based on handcrafted features.** As presented in the main article, the RF and SVM models rely on handcrafted features based on morphological and syntactical attributes. As some features are extracted from only one of the two components of an article, while other features depend on both components, we provide a more detailed list of the extracted features below:

- title features: part-of-speech tags, LIX, RIX, question word counts, punctuation character counts;

- body features: LIX, RIX, CLScore;

| Hyperparameter | Value |
|---|---|
| Number of estimators | 150 |
| Criterion | Entropy |
| Class weights | Balanced |
| Out-of-bag score | True |

Table 3: Optimal hyperparameter configuration for the Random Forest classifier implemented in scikit-learn. Other hyperparameters are left to their default values.

| Hyperparameter | Value |
|---|---|
| Kernel | linear |
| $C$ | 1 |
| Probability | True |

Table 4: Optimal hyperparameter configuration for the SVM classifier implemented in scikit-learn. Other hyperparameters are left to their default values.

- shared features: common noun counts, proper noun counts.

The resulting feature vectors are standardized, before being passed to the classifiers. The Random Forest classifier and the SVM classifier are based on an optimal set of hyperparameters established through validation. These hyperparameters are presented in Table 3 and Table 4, respectively.

**BiLSTM network.** This model uses the Keras tokenizer to extract tokens from titles and bodies. We keep the most frequent 12,000 tokens from the headlines and 25,000 tokens from the contents, for which we extract 300-dimensional FastText embeddings. The titles and contents are processed by two separate BiLSTM branches, and the resulting embeddings are subsequently merged. Each branch is formed of two BiLSTM layers, with 32 units for titles and 64 units for contents. Each branch ends with a global max-pooling layer. The resulting feature vectors are concatenated and passed through two fully-connected layers with 128 and 64 neurons, respectively. Both fully-connected layers have a dropout rate of 0.6. The final layer performs softmax classification. The model is trained for 10 epochs with a mini-batch size of 32 samples using the Adam optimizer (Kingma and Ba, 2017).

**Fine-tuned Ro-BERT.** This approach is based on fine-tuning the Romanian BERT model to generate embeddings of the input text. In our case, the text is given by the concatenated title and body, which are separated by the special separation token. We attach a dropout layer with a rate of 0.2, a dense layer of 128 neurons, and a softmax classification

layer. The modified Ro-BERT model is trained for 10 epochs with a mini-batch size of 4 samples, while using the AdamW optimizer (Loshchilov and Hutter, 2017) with a learning rate of $2 \cdot 10^{-5}$.

**Contrastive Ro-BERT.** We employ the Ro-BERT tokenizer to generate the embeddings for each sample, while separately processing the title and the body. The maximum length of each sequence is set to 256. Before using them in later stages, these embeddings are normalized by subtracting the mean token embedding of the non-padding tokens. Each text and body pair is then used in the proposed loss function. The model is trained for 5 epochs with a mini-batch size of 4 samples, using the Adam optimizer with a fixed learning rate of $10^{-6}$. During inference, we compute the cosine similarity between the title and the body forming a test sample. The resulting similarity value is used to separate each test article into non-clickbait and clickbait, using a threshold of $0.75$.

**Ensemble of the five baselines.** We apply a weighted voting system to determine the label of a sample, based on the outputs of the five individual approaches. Based on preliminary validation experiments, we set the weights as follows: 0.19 for the Random Forest classifier, 0.19 for the SVM classifier, 0.22 for the BiLSTM network, 0.19 for fine-tuned Ro-BERT, and 0.21 for the contrastive Ro-BERT. For the ensemble, we set the clickbait threshold at $0.5$.

## B Appendix: Discussion

Romanian is an Eastern Romance language, being part of a linguistic group that evolved from several dialects of Vulgar Latin which separated from the Western Romance languages between the 5th and the 8th centuries AD. Being surrounded by Slavic neighbors, it has a strong Slavic influence, which makes Romanian a unique language. Moreover, clickbait is often identified via subtle (semantic) differences between title and content, which can easily get lost in translation to more popular languages. To test the ability of multilingual models in Romanian clickbait detection, we fine-tune the multilingual BERT on our dataset, obtaining a macro $F_1$ of 0.6188, which is well below the macro $F_1$ of Ro-BERT, indicating that pre-training on other languages does not bring any benefits to Romanian clickbait detection.

To create clickbait, news writers put their well-trained language skills to use, making clickbait detection a very difficult task. The tricks involved range from simple punctuation tricks to incomplete phrases suggesting a different topic. For instance, the title "The trip to India made Dan Negru very rich" suggests that Dan Negru went to India for some business and made a lot of money, but the content might reveal that Dan Negru became "spiritually rich", which is a completely different thing. In this example, a model would have to figure out that the word "rich" is used with different meanings in the title and the content, respectively. This example illustrates that detecting clickbait is indeed a challenging task.

We consider that all the above aspects combined justify the study of clickbait detection for Romanian.