# OpenReview forum: "A Novel Contrastive Learning Method for Clickbait Detection on RoCliCo: A Romanian Clickbait Corpus of News Articles"
_EMNLP/2023/Conference — EMNLP 2023 Findings_

### Official Review · Reviewer_T8m7 · 2023-07-24

**Soundness:** 2

**Excitement:**

2: Mediocre: This paper makes marginal contributions (vs non-contemporaneous work), so I would rather not see it in the conference.

**Paper Topic And Main Contributions:**

This paper focuses on the clickbait detection task and collects a new Romanian clickbait dataset (low-resource languages) for future research.
The main contributions consist of collecting a new Romanian clickbait dataset and adopting contrastive learning into clickbait detection task.

**Questions For The Authors:**

A. (Line 200-202) Is this phenomenon announced by the website itself or by the author's study? If it is a study, please give an argument, otherwise I think this conclusion is a strong hypothesis lack of justification.
B. The conclusion does not give insight and line332 mentioned that "non-clickbait articles is fairly easy. However, clickbait articles are comparatively",  is it possible that the data set is biased (4,460 non-clickbait samples and 3,453 clickbait samples)?

**Reasons To Accept:**

The main contributions consist of collecting a new Romanian clickbait dataset and adopting contrastive learning into clickbait detection task.

**Reasons To Reject:**

1. The description of the dataset is not specific enough, such as the domains, the average number of sentences of these articles, how many sentences are in the longest article, and how many sentences are in the shortest article.
2. The data collection method, the annotation process lack of details, for example, what is the annotator agreement score?
2. Do not describe the difference between their proposed method and Siamese networks.
3. Do not clarify the difference between fake news detection, satirical news detection and clickbait detection. Maybe the author can explain it in the intro section.

**Reproducibility:**

3: Could reproduce the results with some difficulty. The settings of parameters are underspecified or subjectively determined; the training/evaluation data are not widely available.

**Reviewer Confidence:**

4: Quite sure. I tried to check the important points carefully. It's unlikely, though conceivable, that I missed something that should affect my ratings.

**Typos Grammar Style And Presentation Improvements:**

line 278: for all i = ***, may be more understandable to use "for all $i \in [1,n]$" (Latex)

---

> ### Author Rebuttal · Authors · 2023-08-26
>
> 1. _The description of the dataset is not specific enough, such as the domains, the average number of sentences of these articles, how many sentences are in the longest article, and how many sentences are in the shortest article._
>
> Reply: Regarding the domains, we do not impose any restrictions. We collected samples from multiple domains, including culture, politics, finance, science, technology, sports, celebrities, etc. The distribution of samples across domains is roughly uniform. In lines 177-181, we provided statistics about the number of tokens. As suggested by the reviewer, we also computed the average number of sentences per article, as well as the minimum number of sentences and the maximum number of sentences. The numbers are given below:
>
> | Average | Minimum | Maximum |
> |-----------|-----------|-----------|
> | 28 | 3 | 448|
>
> We will add these statistics in the paper. We thank the reviewer for this recommendation.
>
> 2. _The data collection method, the annotation process lack of details, for example, what is the annotator agreement score?_
>
> Reply: We detailed the annotation process in lines 182-208. At the end of that paragraph, we reported the inter-annotator Kappa agreement score of 0.73. In lines 167-181, we also clearly mentioned the five websites from where the news articles are collected.
>
> 3. _Do not describe the difference between their proposed method and Siamese networks._
>
> Reply: In lines 266-267, we clearly mentioned that we used a Siamese BERT. As mentioned in lines 147-152, the novelty of our approach consists of applying the Siamese BERT on (title, content) pairs, where the title and the content represent a single data sample from our corpus. Related approaches applied siamese networks on (title_i, title_j) pairs, i.e. they consider titles from different data samples. This approach requires hard example mining that results in an expensive training process. In contrast, our approach has the same training complexity as the standard BERT.
>
> 4. _Do not clarify the difference between fake news detection, satirical news detection and clickbait detection. Maybe the author can explain it in the intro section._
>
> Reply: Satirical news form a type of fake news, where the intention of the author is to make the readers laugh. In satirical news, it is clear for the readers that the reported events are fictitious. Fake news represent news that report fake events or facts, where the intention of the author is to deceive the readers into believing that the fake events or facts are true. Clickbait news is a distinct category where the title of news is deceptive with respect to the content. In some clickbait news, the content can present real events and true facts. In other clickbait news, the reported events or facts can be fake. Hence, these three categories (satirical, fake, clickbait) are distinct, although their intersection is clearly not empty. We will add these clarifications in the introduction. We thank the reviewer for the constructive feedback.
>
> 5. _(Line 200-202) Is this phenomenon announced by the website itself or by the author's study? If it is a study, please give an argument, otherwise I think this conclusion is a strong hypothesis lack of justification._
>
> Reply: This is an observation of the authors, definitely not a conclusion. It is just provided as a hint for the annotators. More precisely, in the instructions provided to the annotators, we told the annotators to be extra careful when they have to annotate a news article from the politics, finance, or celebrity categories. However, they were free to judge and assign the clickbait or non-clickbait labels on a case by case basis, as mentioned in lines 202-206. We will reformulate the sentence to clarify this point.
>
> 6. _The conclusion does not give insight and line 332 mentioned that "non-clickbait articles is fairly easy. However, clickbait articles are comparatively", is it possible that the data set is biased (4,460 non-clickbait samples and 3,453 clickbait samples)?_
>
> Reply: We agree that the imbalance nature of the dataset can be the reason behind the reported results. However, testing models on intentionally balanced data does solve the problem in the real world. Instead, models should rather be developed to better cope with the imbalanced task. We will add this clarifying comment in the camera ready.
>
> 7. _Line 278: for all i = ***, may be more understandable to use "for all i ∈ [1, n]" (Latex)_
>
> Reply: We will make this change, as suggested by the reviewer.
>
> 8. _Reproducibility concerns._
>
> Reply: Please note that the collected data and the code to reproduce the baselines are attached as supplementary materials. We will publicly release the code and data to ensure the full reproducibility of our results by readers. We believe this should alleviate all concerns on reproducibility.

---

### Official Review · Reviewer_X7XQ · 2023-08-04

**Soundness:** 3

**Excitement:**

3: Ambivalent: It has merits (e.g., it reports state-of-the-art results, the idea is nice), but there are key weaknesses (e.g., it describes incremental work), and it can significantly benefit from another round of revision. However, I won't object to accepting it if my co-reviewers champion it.

**Paper Topic And Main Contributions:**

This paper presents the authors' experience in Clickbait Detection, introducing RoCliCo, the first public corpus for Romanian clickbait detection. They train and evaluate four models and an ensemble of these models to establish competitive baselines for future research on their corpus. The authors propose a novel approach using a BERT model based on contrastive learning, which leverages cosine similarity between the title and content to detect clickbait articles. The empirical results indicate significant potential for future research on the Romanian clickbait detection task. The availability of RoCliCo opens up opportunities for collaboration and advancement in clickbait detection methods for the Romanian language.

**Questions For The Authors:**

- Is there any related work for Clickbait Detection? How are their methods different from your work?

- Please provide detailed information on the qualifications of the annotators and the payment process for the labeling task to ensure transparency and reliability.

- The rationale behind separating the news platforms between the training and test sets (as discussed in Section 3) should be clearly explained to justify the authors' approach.

- Table 1 should be appropriately placed in the text, precisely where it is mentioned, to enhance readability and facilitate easy reference.

- To demonstrate the dataset's utility, it is essential to compare it with popular datasets to highlight the unique dimensions it can contribute to various research areas.

- While the authors have presented a comprehensive overview of the Clickbait Detection task, including additional information on the linguistic aspects of the task would be valuable. This addition would provide readers with a deeper understanding of the study's theoretical foundations and the methodologies used to achieve the results.

- To ascertain the significance of the reported results, the authors should provide statistical analysis. This will offer readers a more accurate understanding of the effectiveness of the methods employed.

- The manuscript lacks mention of hyperparameters or other model parameters. It is crucial to explain how these parameters are optimized to ensure reproducibility and enable others to apply the methodology effectively.

**Reasons To Accept:**

- The process of creating the dataset is thoroughly explained.
- A dataset of Clickbait in Romanian with comprehensive annotations holds potential value for researchers.
- The authors conduct multiple experiments to validate the efficacy of both handcrafted and deep models as baseline methods on the dataset.
- Reporting the performance of various commonly used models on this dataset serves as a valuable reference for future benchmarking efforts.

**Reasons To Reject:**

- Please provide detailed information on the qualifications of the annotators and the payment process for the labeling task to ensure transparency and reliability.

- The rationale behind separating the news platforms between the training and test sets (as discussed in Section 3) should be clearly explained to justify the authors' approach.

- Table 1 should be appropriately placed in the text, precisely where it is mentioned, to enhance readability and facilitate easy reference.

- To demonstrate the dataset's utility, it is essential to compare it with popular datasets to highlight the unique dimensions it can contribute to various research areas.

- While the authors have presented a comprehensive overview of the Clickbait Detection task, including additional information on the linguistic aspects of the task would be valuable. This addition would provide readers with a deeper understanding of the study's theoretical foundations and the methodologies used to achieve the results.

- To ascertain the significance of the reported results, the authors should provide statistical analysis. This will offer readers a more accurate understanding of the effectiveness of the methods employed.

- The manuscript lacks mention of hyperparameters or other model parameters. It is crucial to explain how these parameters are optimized to ensure reproducibility and enable others to apply the methodology effectively.

**Reproducibility:**

3: Could reproduce the results with some difficulty. The settings of parameters are underspecified or subjectively determined; the training/evaluation data are not widely available.

**Reviewer Confidence:**

5: Positive that my evaluation is correct. I read the paper very carefully and I am very familiar with related work.

**Typos Grammar Style And Presentation Improvements:**

- To make the analysis of the results more transparent, authors should present their results in several metrics and as a 4-digit decimal (ex: 0.81xx).

---

> ### Author Rebuttal · Authors · 2023-08-26
>
> 1. _Please provide detailed information on the qualifications of the annotators and the payment process for the labeling task to ensure transparency and reliability._
>
> Reply: The annotators are native Romanian speakers, who graduated from their bachelor studies at Romanian universities. We checked their reading comprehension skills prior to enrolment. Then, they received instructions on how to perform the annotations. As mentioned in the paper, the annotators are volunteers who agreed to assign the labels for free.
>
> 2. _The rationale behind separating the news platforms between the training and test sets (as discussed in Section 3) should be clearly explained to justify the authors' approach._
>
> Reply: Without source separation between training and test, a model that overfits to certain source-specific aspects that are not related to clickbait (e.g. author style, preference towards certain subjects, etc.) will reach high scores on the test. However, these scores are unlikely to represent the actual performance of the model in a real-world scenario, where news can come from different sources without annotated samples to be used for training. We consider that a more realistic evaluation is to separate the platforms. In the proposed setting, models that learn patterns related to author style, preferred themes, and such will not be able to capitalize on features unrelated to clickbait detection. We will add these clarifying comments in the camera ready. We thank the reviewer for the constructive feedback.
>
> 3. _Table 1 should be appropriately placed in the text, precisely where it is mentioned, to enhance readability and facilitate easy reference._
>
> Reply: We will move the table just before the paragraph, as suggested by the review. We thank the reviewer for this observation.
>
> 4. _To demonstrate the dataset's utility, it is essential to compare it with popular datasets to highlight the unique dimensions it can contribute to various research areas._
>
> Reply: The relation to existing clickbait corpora in other languages is detailed in Section 2 (please see lines 106-129). In terms of size, our corpus is in the same range as the other corpora for low-resource languages. However, most of the other corpora are based on distant supervision, labeling the news as clickbait based on the source platforms. In contrast, our corpus is based on manual annotations. We observed that news articles from a certain source platform cannot be placed in the same bucket, although it is true that different platforms use clickbait in different percentages (between 17% and 71%). To address this concern, we will add a table with comparative statistics between related corpora in the camera ready. We thank the reviewer for the observation.
>
> 5. _While the authors have presented a comprehensive overview of the Clickbait Detection task, including additional information on the linguistic aspects of the task would be valuable. This addition would provide readers with a deeper understanding of the study's theoretical foundations and the methodologies used to achieve the results._
>
> Reply: Certainly, news writers use their language skills to create clickbait. The tricks involved range from simple punctuation tricks (e.g. the title is “Who is the person that killed three people on Thursday” without a question mark, which indicates that the content is going to reveal this person’s name. However, the content mentions that the chief of police declared that “we need to find out who is the person that killed three people on Thursday and ensure this will not happen again”, without actually naming the person) to incomplete phrases suggesting a different topic (e.g. the title is “The trip to India made Dan Negru very rich”, suggesting that Dan Negru went to India for some business and made a lot of money, but the content reveals that Dan Negru became “spiritually rich”, which is completely different). In the latter example, a model would have to figure out that the word “rich” was used with different meanings in the title and the content, respectively. This example shows that detecting clickbait is indeed a challenging task. We will add this discussion in the paper.
>
> 6. _To ascertain the significance of the reported results, the authors should provide statistical analysis. This will offer readers a more accurate understanding of the effectiveness of the methods employed._
>
> Reply: Since our main methodological contribution is the contrastive learning approach, we evaluated the statistical significance of the proposed contrastive Ro-BERT with respect to the fine-tuned Ro-BERT using a paired McNemar’s test. The test indicates that the performance improvement brought by our contrastive learning method is statistically significant, with a p-value of 0.05. We will add this result in the camera ready. We thank the reviewer for the constructive comment.
>
> 7. _The manuscript lacks mention of hyperparameters or other model parameters. It is crucial to explain how these parameters are optimized to ensure reproducibility and enable others to apply the methodology effectively._
>
> Reply: Please note that the collected data and the code to reproduce the baselines are attached as supplementary materials. We will publicly release the code and data to ensure the full reproducibility of our results by readers. We believe this should alleviate all concerns on reproducibility.
>
> 8. _Is there any related work for Clickbait Detection? How are their methods different from your work?_
>
> Reply: To the best of our knowledge, there is no clickbait detection corpus for the Romanian language. As mentioned in Section 2 (lines 106-129), there are related corpora for other low-resource languages. In terms of the proposed approach based on contrastive learning, there are only two studies based on contrastive learning for clickbait news detection, as discussed in lines 135-152 for Section 2. Related approaches applied contrastive learning on (title_i, title_j) pairs, i.e. they consider titles from different data samples. This approach requires hard example mining that results in an expensive training process. Different from these works, we use (title_i, content_i) pairs, where the title_i and the content_i represent the i-th data sample from the training set. In our approach, clickbait samples are regarded as negative pairs, and non-clickbait samples are regarded as positive pairs. Hence, there is no hard sample mining involved. Our model learns a metric space between titles and contents, while existing contrastive learning methods learn a metric space between titles, which is fundamentally different. We will add these comments in the paper to better explain how our approach is different.
>
> 9. _To make the analysis of the results more transparent, authors should present their results in several metrics and as a 4-digit decimal (ex: 0.81xx)._
>
> Reply: We thank the reviewer for this recommendation. We will update all results in Table 2 accordingly. Please note that we reported results with 4-digit decimals for two extra baselines in the rebuttal to Reviewer DCj2.

---

### Official Review · Reviewer_dgex · 2023-08-07

**Paper Topic And Main Contributions:** Gather qnd curate a click bait classi…
**Typos Grammar Style And Presentation Improvements:** Some simple edits are necessary but n…
**Soundness:** 4

**Excitement:**

3: Ambivalent: It has merits (e.g., it reports state-of-the-art results, the idea is nice), but there are key weaknesses (e.g., it describes incremental work), and it can significantly benefit from another round of revision. However, I won't object to accepting it if my co-reviewers champion it.

**Missing References:**

Other click bait datasets across languages and a side by side comparison.

**Questions For The Authors:**

Why is it important to have this dataset in Romanian? How is it different than other languages?

**Reasons To Accept:**

It's a novel dataset and has well established labeling, reliability metrics, and

**Reasons To Reject:**

For classification tasks it would be good to see a PR curve to evaluate overall performance rather than just mean metrics.

**Reproducibility:**

4: Could mostly reproduce the results, but there may be some variation because of sample variance or minor variations in their interpretation of the protocol or method.

**Reviewer Confidence:**

4: Quite sure. I tried to check the important points carefully. It's unlikely, though conceivable, that I missed something that should affect my ratings.

---

> ### Author Rebuttal · Authors · 2023-08-26
>
> 1. _For classification tasks it would be good to see a PR curve to evaluate overall performance rather than just mean metrics._
>
> Reply: As suggested, we computed the PR curve for a number of models. We added the figure at: https://postlmg.cc/w1qcXYR8. We will include the PR curves of all models in the final camera ready version. We thank the reviewer for the constructive feedback.
>
> 2. _Why is it important to have this dataset in Romanian? How is it different from other languages?_
>
> Reply: From our evaluation of multiple news sources (beyond those included in RoClico), all Romanian news sources seem to use clickbait titles to some extent. This is likely because there are no subscription based publications in Romania, and the news publishers have to rely on advertisement to obtain earnings. However, clickbait is often identified via subtle (semantic) differences between title and content, which can easily get lost in translation. Romanian is an Eastern Romance language, being part of a linguistic group that evolved from several dialects of Vulgar Latin which separated from the Western Romance languages between the 5th and the 8th centuries AD. Being surrounded by Slavic neighbors, it has a strong Slavic influence, which makes Romanian a unique language. We fine-tuned multilingual BERT on our dataset and obtained a macro F1 of 0.6188, which is well below the macro F1 of Ro-BERT, indicating that pre-training on other languages does not bring any benefits to Romanian clickbait detection. We consider that all these aspects combined justify the study of clickbait detection for Romanian.
>
> 3. _Other click bait datasets across languages and a side by side comparison._
>
> Reply: In lines 106 to 129, we relate to clickbait datasets in other languages, indicating that our corpus has a similar size to existing corpora. For the camera ready, we will add a table with comparative statistics between clickbait datasets. We thank the reviewer for this recommendation.

---

### Official Review · Reviewer_DCj2 · 2023-08-11

**Soundness:** 3

**Excitement:**

3: Ambivalent: It has merits (e.g., it reports state-of-the-art results, the idea is nice), but there are key weaknesses (e.g., it describes incremental work), and it can significantly benefit from another round of revision. However, I won't object to accepting it if my co-reviewers champion it.

**Paper Topic And Main Contributions:**

The authors present RoCliCo, a corpus that is a collection of over 7,900 news samples in Romanian that are manually annotated with clickbait and non-clickbait labels. It is the first publicly available corpus for clickbait detection in Romanian news, and it provides a valuable resource for researchers and practitioners to develop and evaluate machine learning models for this task. Experiments with a few machine learning methods to establish competitive baselines, which can serve as a starting point for future research on clickbait detection in Romanian.

**Questions For The Authors:**

Can you provide more details on the criteria used for selecting the five Romanian news websites included in the corpus? Were there any specific reasons for choosing these websites over others?

How did you ensure the quality and consistency of the annotations provided by the volunteers? Did you conduct any inter-annotator agreement analysis or provide any guidelines for the annotators?

As the clickbait detection task is quite similar to the natural language inference task (NLI), why do the authors not pick some existing NLI models for comparison?

Section 3: In the last paragraph, “… the test data represents a more natural distribution, …”, what is “natural distribution”? Is imbalance data the same as “natural distribution”?

Section 4: In the 1st paragraph, “… based on part-of-speech, scores, and punctuation patterns.”, what do the scores represent?

Section 4: In the 2nd paragraph, “Each input is first preprocessed by a tokenizer.”, which tokenizer do the authors employ? NLTK, Gensim, or StanfordCoreNLP?

Section 4: In the 2nd paragraph, “The resulting embeddings are passed as input to the neural network.”, what neural network is chosen? MLP, CNN, or Transformer?

Section 4: In the 3rd paragraph, “The generated Ro-BERT-based embeddings are passed through a dropout layer, a fully-connected layer, and a classification layer.”, as the former sentence mentioned [SEP], does here the embedding denote [CLS] embedding?

**Reasons To Accept:**

This paper is really well-motivated and well-written.

The RoCliCo enriches the low-resource natural language corpora and clickbait detection.

The selected baseline models show that Romanian clickbait detection is a challenging task.

**Reasons To Reject:**

The number and type of baseline models in the Experiments section are too few, only 4.

The size of RoCliCo is relatively small since the majority of non-clickbait samples can be collected from some authoritative news website.

Though Cohen’s kappa coefficient among annotators is 0.73, the labeling process lack of cross-check, which can introduce biases and errors in the labeling results.

**Reproducibility:**

4: Could mostly reproduce the results, but there may be some variation because of sample variance or minor variations in their interpretation of the protocol or method.

**Reviewer Confidence:**

4: Quite sure. I tried to check the important points carefully. It's unlikely, though conceivable, that I missed something that should affect my ratings.

---

> ### Author Rebuttal · Authors · 2023-08-26
>
> 1. _The number and type of baseline models in the Experiments section are too few, only 4._
>
> Reply: To address this point, we conducted additional experiments with a shallow model (SVM) based on the same features as the Random Forest, and a multilingual BERT that is first fine-tuned on XNLI, then fine-tuned on RoCliCo. The results of these models are presented below:
>
> | Model | Precision (clickbait) | Recall (clickbait) | F1 (clickbait) | Precision (non-clickbait) | Recall (non-clickbait) | F1 (non-clickbait) | Macro F1 |
> |-----------|-----------|-----------|-----------|-----------|-----------|-----------|-----------|
> | SVM | 0.7142 | 0.5780 | 0.6389 | 0.9245 | 0.9571 | 0.94056 | 0.7897 |
> | Multilingual BERT | 0.4731 | 0.2528 | 0.3295 | 0.8717 | 0.9474 | 0.9080 | 0.6188 |
>
> We will include these additional models in Table 2 from the camera ready. We thank the reviewer for the constructive feedback.
>
> 2. _As the clickbait detection task is quite similar to the natural language inference task (NLI), why do the authors not pick some existing NLI models for comparison?_
>
> Reply: To the best of our knowledge, there is no NLI dataset and model for the Romanian language. Hence, the only option is to take a multilingual model pre-trained on XNLI (although XNLI does not contain sample translations to Romanian), and fine-tune it on our dataset. As earlier mentioned, a fine-tuned multilingual BERT obtains a macro F1 of 0.6188, which is below the macro F1 of Ro-BERT (please see Table 2). This confirms that the language-specific Ro-BERT is able to obtain better results. We thank the reviewer for the constructive comment.
>
> 3. _The size of RoCliCo is relatively small since the majority of non-clickbait samples can be collected from some authoritative news website._
>
> Reply: When we selected our sources, we initially considered protv.ro and digi24.ro as credible news sources. However, after the labeling process, we found clickbait news even from these sources (the minimum clickbait ratio per source is 17%). Unfortunately, all Romanian news sources seem to use clickbait titles to some extent. This is likely because there are no subscription-based publications in Romania, and the news publishers have to rely on advertisement to obtain earnings. We will clarify this in the camera ready.
>
> 4. _Though Cohen’s kappa coefficient among annotators is 0.73, the labeling process lack of cross-check, which can introduce biases and errors in the labeling results._
>
> Reply: The authors checked the assigned labels for a subset of 200 samples and confirmed the consistency of the labels. The final label for a sample is decided based on the majority vote of the three annotators, which reduces labeling errors by a large extent.
>
> 5. _Can you provide more details on the criteria used for selecting the five Romanian news websites included in the corpus? Were there any specific reasons for choosing these websites over others?_
>
> Reply: The chosen five Romanian news websites were selected based on their popularity among Romanians, as well as their reputation as credible or less credible sources. As protv.ro and digi24.ro sources cover a wide range of news categories and are considered as credible news sources by the population. In contrast, wowbiz.ro and cancan.ro tend to concentrate more on celebrity news, and use clickbait to attract readers. However, our annotation process revealed that these websites contain both clickbait and non-clickbait articles, though the percentage of clickbait news per source can differ significantly, between 17% and 71%.
>
> 6. _How did you ensure the quality and consistency of the annotations provided by the volunteers? Did you conduct any inter-annotator agreement analysis or provide any guidelines for the annotators?_
>
> Reply: Yes, we measured the agreement among annotators and obtained a Kappa coefficient of 0.73, which confirms that the inter-annotator agreement is substantial. Each annotator was asked to read a instructive document written in Romanian, where the authors provided definitions, guidelines and 5 examples of clickbait and non-clickbait news articles.
>
> 7. _Section 3: In the last paragraph, “… the test data represents a more natural distribution, …”, what is “natural distribution”? Is imbalance data the same as “natural distribution”?_
>
> Reply: We used two news sources, Cancan and WowBiz, that are likely to have clickbait news and only one source that is likely to have credible news, essentially skewing the distribution towards having more clickbait news. This led to a roughly balanced training set. For the test set, we used one credible and one less credible source, and the resulting distribution is indeed imbalanced. We consider the latter distribution to be closer to the natural one, since there was no intervention from our side to collect more clickbait news for the test set.
>
> 8. _Section 4: In the 1st paragraph, “… based on part-of-speech, scores, and punctuation patterns.”, what do the scores represent?_
>
> Reply: As mentioned on line 227, the features are taken from Coste et al. (2020). In this paper, the authors use the CLScore [1], LIX index [2] and RIX index [2]. To make our paper more self-contained, we will clarify this point in the camera ready version. We thank the reviewer for this suggestion.
>
> 9. _Section 4: In the 2nd paragraph, “Each input is first preprocessed by a tokenizer.”, which tokenizer do the authors employ? NLTK, Gensim, or StanfordCoreNLP?_
>
> Reply: We use the same tokenizer mentioned on line 234, namely Stanza (Qi 234 et al., 2020). We will add this mention in the camera ready to improve clarity. We thank the reviewer for the constructive feedback.
>
> 10. _Section 4: In the 2nd paragraph, “The resulting embeddings are passed as input to the neural network.”, what neural network is chosen? MLP, CNN, or Transformer?_
>
> Reply: This paragraph, from lines 239 to 250, is for the BiLSTM network. Hence, the neural network is a BiLSTM (Bidirectional Long Short-Term Memory) network. We will clarify this aspect in the camera ready. We thank the reviewer for the constructive comment.
>
> 11. _Section 4: In the 3rd paragraph, “The generated Ro-BERT-based embeddings are passed through a dropout layer, a fully-connected layer, and a classification layer.”, as the former sentence mentioned [SEP], does the embedding denote [CLS] embedding?_
>
> Reply: Yes, this paragraph refers to the [CLS] embeddings. We will clarify this point in the paper. Thank you for the clarity concerns mentioned above. These will definitely improve the clarity of our paper.
>
> **References:**
>
> [1] M. Coleman and T.L. Liau, “A computer readability formula designed for machine scoring, in Journal of Applied Psychology,” 1975, Vol. 60. pp. 283-284.
>
> [2] J. Anderson, “Lix and Rix: Variations on a Little-known Readability Index,” in Journal of Reading 26 (6), pp. 490-496, 1983.

---

### Meta-Review · Area_Chair_B5Bg · 2023-09-21

**Recommendation:** 4

**Metareview:**

This study proposed clickbait detection dataset in Romanian, which is first of its kind. The dataset is comprised with 7,913 news samples, which are manually annotated with clickbait and non-clickbait labels.

Reviewers provided insightful feedback. For instance, Reviewer DCj2 pointed out an imbalanced distribution in the dataset (e.g., a majority of non-clickbait samples). This should be clarified to improve the quality of the paper. This reviewer also expressed concerns about the baseline experiments. Since this study aims to serve as a baseline for future work, it is important to present additional results using different models. The selection of these models could be based on those that are widely used and demonstrate superior performance in related studies.

Reviewer X7XQ suggested comparing the dataset with other popular datasets. It will help readers understand the utility and significance of the presented dataset.

---

### Decision · Program_Chairs · 2023-10-07

**Decision:**

Accept-Findings

**Comment:**

This study proposed clickbait detection dataset in Romanian, which is first of its kind. The dataset is comprised with 7,913 news samples, which are manually annotated with clickbait and non-clickbait labels.

Reviewers provided insightful feedback. For instance, Reviewer DCj2 pointed out an imbalanced distribution in the dataset (e.g., a majority of non-clickbait samples). This should be clarified to improve the quality of the paper. This reviewer also expressed concerns about the baseline experiments. Since this study aims to serve as a baseline for future work, it is important to present additional results using different models. The selection of these models could be based on those that are widely used and demonstrate superior performance in related studies.

Reviewer X7XQ suggested comparing the dataset with other popular datasets. It will help readers understand the utility and significance of the presented dataset.